# Gait Detection from a Wrist-Worn Sensor Using Machine Learning Methods: A Daily Living Study in Older Adults and People with Parkinson’s Disease

**DOI:** 10.3390/s22187094

**Published:** 2022-09-19

**Authors:** Yonatan E. Brand, Dafna Schwartz, Eran Gazit, Aron S. Buchman, Ran Gilad-Bachrach, Jeffrey M. Hausdorff

**Affiliations:** 1Center for the Study of Movement, Cognition and Mobility, Neurological Institute, Tel Aviv Sourasky Medical Center, Tel Aviv 6492416, Israel; 2Sagol School of Neuroscience, Tel Aviv University, Tel Aviv 6997801, Israel; 3Department of Biomedical Engineering, Faculty of Engineering, Tel Aviv University, Tel Aviv 6997801, Israel; 4Rush Alzheimer’s Disease Center, Department of Neurological Sciences, Rush University Medical Center, Chicago, IL 60612, USA; 5Edmond J. Safra Center for Bioinformatics, Tel-Aviv University, Tel Aviv 6997801, Israel; 6Rush Alzheimer’s Disease Center and Department of Orthopedic Surgery, Rush University, Chicago, IL 60612, USA; 7Department of Physical Therapy, Sackler Faculty of Medicine, Tel Aviv University, Tel Aviv 6997801, Israel

**Keywords:** gait, Parkinson’s disease, machine learning, inertial measurement unit (IMU), wrist, accelerometer

## Abstract

Remote assessment of the gait of older adults (OAs) during daily living using wrist-worn sensors has the potential to augment clinical care and mobility research. However, hand movements can degrade gait detection from wrist-sensor recordings. To address this challenge, we developed an anomaly detection algorithm and compared its performance to four previously published gait detection algorithms. Multiday accelerometer recordings from a wrist-worn and lower-back sensor (i.e., the “gold-standard” reference) were obtained in 30 OAs, 60% with Parkinson’s disease (PD). The area under the receiver operator curve (AUC) and the area under the precision–recall curve (AUPRC) were used to evaluate the performance of the algorithms. The anomaly detection algorithm obtained AUCs of 0.80 and 0.74 for OAs and PD, respectively, but AUPRCs of 0.23 and 0.31 for OAs and PD, respectively. The best performing detection algorithm, a deep convolutional neural network (DCNN), exhibited high AUCs (i.e., 0.94 for OAs and 0.89 for PD) but lower AUPRCs (i.e., 0.66 for OAs and 0.60 for PD), indicating trade-offs between precision and recall. When choosing a classification threshold of 0.9 (i.e., opting for high precision) for the DCNN algorithm, strong correlations (r > 0.8) were observed between daily living walking time estimates based on the lower-back (reference) sensor and the wrist sensor. Further, gait quality measures were significantly different in OAs and PD compared to healthy adults. These results demonstrate that daily living gait can be quantified using a wrist-worn sensor.

## 1. Introduction

Wrist-worn sensors are widely used in research studies and clinical trials to quantify movement. Varied behaviors of daily living, including sleep and circadian metrics, and the diverse facets of physical activity can be extracted and quantified from recordings obtained from an inertial measurement unit (IMU) such as an accelerometer and gyroscope. These metrics have been shown to predict morbidity, mortality, cognitive decline, and the development of dementia [1,2,3] and have also been employed for the early detection and monitoring of neurodegenerative disorders that commonly manifest impaired gait (e.g., Alzheimer’s and Parkinson’s disease (PD)) [4,5,6].

Many previous studies using wrist-worn sensors concentrated on quantifying the level of physical activity, predominantly a reflection of the total amount of daily walking [7,8,9]. As gait is a relatively complex behavior, recent efforts have focused on developing algorithms that can extract additional facets of gait quality (e.g., gait speed, cadence, and step regularity) from these same recordings to more fully characterize gait function [10,11,12]. Recent works suggest that quantifying more diverse facets of gait may improve predictions of specific adverse health outcomes [13,14]. Moreover, multiday recordings of gait during daily living may also enhance functional monitoring and the prediction of adverse outcomes such as falls, as compared to short assessments of walking obtained within or outside the lab setting [15,16,17,18,19,20].

To extract measures of gait quality and quantity from acceleration recordings of daily living, it is first necessary to detect the multiple bouts of gait that are contained in a multiday recording. However, reliable detection of walking from wrist-sensor data is challenging due to the many degrees of freedom of wrist movement and because wrist movements may occur in the absence of walking (e.g., when washing dishes or brushing teeth). In other words, the continuous acceleration signals obtained from a wrist sensor do not, a priori, contain reliable labels of gait/non-gait behaviors during daily living. To circumvent this limitation, many previous studies on gait have placed sensors on the lower back or on a lower extremity [4,21,22,23], where it is much easier to reliably identify the bouts of gait.

Existing gait detection algorithms distinguish gait from other behaviors using features from the time and frequency domain, either individually or combined, using conventional statistics [12,24,25]. Once these features are extracted, they can be used as the input for a machine learning model to automatically detect bouts of gait. These include both supervised [26,27] and unsupervised [28] methods. More recently, deep learning techniques have been applied to human activity recognition (HAR) tasks to classify human activity (e.g., walking, sitting, and driving) [29,30,31,32]. Deep learning models can deal with complex and large datasets and offer an approach for automatic gait detection from wrist recordings of daily living [31].

Only a few studies have developed and validated gait detection algorithms using wrist-sensor recordings of daily living [11,26,27,28,33,34]. One of the challenges of developing daily living gait detection algorithms is that walking during daily living occurs for only a small portion of the overall recording, as non-gait behaviors constitute the bulk of the overall recording. Thus, false-positive predicted bouts of gait occur frequently when analyzing daily living recordings. Hence, there is a significant trade-off between precision (i.e., a positive predictive value) and sensitivity (i.e., the percent of correctly detected gait bouts, also referred to as recall) of the algorithm. Thus, focusing on algorithm performance statistics, such as accuracy and specificity, without assessing precision, as has been done in the past, [28,30,31], will not provide an adequate assessment of the algorithm’s performance. 

Many prior algorithms have been developed for healthy, younger adults. Gait detection is more difficult in OAs who commonly manifest diverse gait impairments even in the absence of specific diseases. Aging is often accompanied by reduced frequency of gait bouts, and the walking patterns in OAs are less periodic and consistent, increasing the likelihood of false-negative detections of gait. In addition, the presence of arm tremor or reduced arm swing during walking [35] may reduce the strength of the signal obtained from the wrist, making automatic detection of gait more difficult. 

The use of unsupervised anomaly detection algorithms has become a useful approach in cases where labeling is not feasible and examples of one class are sparse [36]. Therefore, it seems reasonable to adopt this approach for the detection of gait from wrist-worn sensors in OAs with and without overt neurological diseases such as Parkinson’s disease (PD). It is possible to address anomaly detection as a one-class classification algorithm that first learns a model of one class during training (which can be referred to as the normal data in classical anomaly detection tasks) and then at the inference time computes an anomaly score of unseen data samples under the learned model [37,38]. Data samples are flagged as belonging to the learned model or not by comparing the anomaly score to a learned decision boundary. In recent years, the practice of normalizing flows, which are deep neural networks based on density estimators, has shown promising performance in different unsupervised anomaly detection tasks [38,39,40,41]. 

The present study adopted an anomaly detection algorithm to detect gait from wrist-worn 3D accelerometer recordings of daily living. We first trained and tested our model using accelerometer recordings from healthy young adults (HYAs) to examine the feasibility of our model. Then, we trained and tested the model using wrist recordings from OAs with and without PD. We compared the performance of the anomaly detection approach to four previously published gait detection algorithms. We extracted measures of gait quantity and gait quality from the best model and compared its performance to those obtained via a sensor worn on the lower back (a form of “gold-standard” reference).

## 2. Materials and Methods

### 2.1. Participants

Wrist-worn recordings of HYAs were obtained from the publicly available PhysioNet dataset [42,43], which included 32 healthy participants (39 ± 9 years, 19 women) who carried out various activities of daily living. The older adult participants were recruited from the Center for the Study of Movement, Cognition, and Mobility (CMCM) at the Neurological Institute in Tel Aviv Sourasky Medical Center, Israel, using a convenience sample of 12 community older adults (OAs) and 18 individuals with PD. The diagnosis of PD was obtained by a movement disorders specialist.

To characterize the OAs and the people with PD, several widely used measures were applied. In the lab, all subjects performed a 2 min walk test (2MWT) while walking at their comfortable speed along a 25 m corridor. A Zeno Walkway Gait Analysis System (Protokinetics LLC, Havertown, PA, USA) with a length of 7.92 m was placed in the middle of the walkway and used to measure the subject’s walking speed. The timed up and go test (TUG) measures the time required to stand up from a seated position, walk 3 m, turn around, and return to a seated position. TUG times above 13.5 s suggest the presence of gait impairments and are associated with an increased risk of falls [44]. The Montreal Cognitive Assessment (MOCA) was used as a general measure of cognitive function. Hoehn and Yahr staging (performed by a movement disorders clinician) was used to describe disease severity among the people with PD.

### 2.2. Wearable Sensors

In the HYA dataset, accelerometry data were collected simultaneously at four body locations—left wrist, left hip, left ankle, and right ankle—at a sampling frequency of 100 Hz (ActiGraph GT3X+, Pensacola, FL, USA). We used only the left wrist data, as we aimed to detect gait from a wrist-worn sensor. The data included labels of activity types performed for each time point in the data collection.

The OAs and people with PD were asked to wear a tri-axial accelerometer on their wrist for up to 10 days, at a sampling frequency of 25 Hz (Garmin, Olathe, KS, USA). The OA participants wore the sensor on their nondominant hand, and the PD participants wore it on their impaired hand (with the idea that the more impaired hand would roughly behave similarly to the nondominant hand and be less involved with non-gait activities). The raw data were uploaded online to cloud services. An additional sensor was placed on the lower back at the level of the fifth lumbar vertebrae (roughly near the center of mass) and the 3D accelerometer signals were sampled at 100 Hz (Axivity AX3, York, UK). Data from the lower back were stored locally and uploaded offline to the lab computers. 

Labeling of the OA and PD datasets to gait and non-gait segments was conducted based on “gold-standard” annotations from the lower-back sensor [4,21,23,45]. The signal from the lower back was less noisy than the wrist and had validated algorithms for detecting gait [46,47]. We used nearest-neighbor interpolation to downsample the annotations from the lower back to match the wrist-sensor sampling rate. We considered only walking bouts with a duration of at least 6 s as “gait”, since shorter duration bouts are extremely difficult to reliably detect, even from a lower-back sensor, and since most gait quality measures cannot be applied to very short walking bouts [48].

### 2.3. Gait Detection Algorithms

In this study, we used both supervised and unsupervised algorithms for gait detection. We split the data into training and testing sets in the supervised models. For the HYA dataset, we used a 70/30 split validation. For the OAs and PD, we used a five-fold cross-validation to split the daily living data into the testing and training sets. Each fold consisted of data from different participants (i.e., the model did not train and test on data from the same participant in each fold) to minimize overfitting of the model. The models were trained and evaluated on each cohort (i.e., HYAs, OAs, and PD) separately, for customizing and optimizing the classifier to the characteristics of each cohort. The different folds included similar percentages of gait, such that they were relatively balanced (15% ± 4.8% for the OAs and 13% ± 2.7% for the PD group).

#### 2.3.1. Anomaly Detection 

We evaluated here, for the first time, an approach of unsupervised anomaly detection for the task of gait detection. We applied the algorithm to the HYA cohort as an initial feasibility test, which allowed us to examine the performance on a simpler and cleaner dataset, and then we applied it to the more complex cohorts of PD and OAs. We addressed the gait detection problem as a one-class classification problem in which the neural network was trained using samples from one class only (gait/non-gait), and this class was considered the “normal” class. The dataset was split into a training set, which consisted of 90% of the “normal” class data, and a validation set, which consisted of the remaining 10% of that “normal” class used in inference time, and a test set which consisted of all the data from the second class (i.e., the “abnormal” class). We examined the model both when gait data were considered normal and used for training, and when non-gait data were considered normal and used for training. At the inference time, to decide if a sample belonged to the “normal” class (i.e., the class that the model was trained on) or to the “abnormal” class (i.e., that the model was not exposed to during training), an anomaly score was calculated, and using a threshold, the samples were classified so that samples that obtained a higher anomaly score than the threshold were considered abnormal, and a sample that obtained a lower score than the threshold was considered as normal. We based our anomaly detection model on the masked autoregressive flow (MAF), which is a type of normalizing flow suitable for density estimation [49]. 

Normalizing flows are neural networks that are used to learn bijective transformations between data distributions and well-defined base densities. The bijectivity is ensured by stacking layers of affine transforms that are fixed or autoregressive. Under this invertibility assumption, the learned density $p\left(x\right)$ can be calculated as:
(1)$$\;\;p\left(x\right)=\pi_{u}\left(f^{-1}\left(x\right)\right)\left|\det\left(\frac{\partial f^{-1}}{\partial x}\right)\right|$$
where $\pi_{u}$ is the well-defined base density, which is usually chosen to be a standard Gaussian density; f is the invertible differential transformation. The MAF model uses a stack of MADE layers, which are a type of autoregressive flow [50]. Our architecture consisted of a stack of 5 MADE layers. The goal during training was to maximize the likelihood or, equivalently, the log-likelihood of the training data with respect to the network parameters. Low-activity segments were removed from the accelerometer signal; a moving window STD was used to remove windows with a low STD (i.e., low activity). We used moving windows with a length of 3 s and set the STD threshold to 0.1. After, principal component analysis (PCA) was conducted for dimensionality reduction. The signals were then split into sliding windows at a fixed width of 6 s with an overlap of 50%, and the DC offset was removed by subtracting the mean value in each window on the three axes. We calculated the mean and standard deviation (STD) of the training set and used it to normalize the training set, validation set, and testing set. Hyperparameters were found for each model of the three cohorts through a set of distinct experiments. Finally, we used an ADAM optimizer with an exponential decay learning rate scheduler with an initial value of 0.01 for the HYA, OA, and PD datasets and an $l_{2}$ regularization of 0.01 for the PD and OA datasets and 0.001 for the HYA dataset. At the inference time, the log-likelihoods of the data samples from the validation set and testing set were computed and used to classify the data according to a selected likelihood threshold. 

#### 2.3.2. Feature-Based Model

Another approach for gait detection is a feature-based (FB) approach. In this method of gait detection, features are extracted from time windows of fixed width for classification [12,24,25]. These features are usually based on the time and frequency domain, either individually or combined. Once features are extracted, we can define thresholds to predict if the window includes gait. In this study, we implemented an FB model published by Keren et al. [24] (Figure 1) for gait detection using a wrist-worn sensor. The original model was developed and validated in an HYA group; here, we applied it to the OAs and people with PD.

#### 2.3.3. Autoregressive Infinite Hidden Markov Model (AR–iHMM)

Classical feature-based models have some challenges when using windows-based features. First, gait features vary between different windows, even in the same participant, depending on the hour of the day, environment, and medication status. Further, extracting frequency domain features relies on the assumption that the window is stationary, which is usually not the case in windows with a fixed width. These problems motivated Raykov et al. [28] to develop a data-driven approach, which did not require the use of windows. In their model, data were automatically segmented into states with variable time lengths. Each state represents a different gait and non-gait pattern. This unsupervised method combines an autoregressive (AR) process, used for parametric estimation of the frequency spectrum, with a hidden Markov model (HMM) used to model the sequence of the different states represented in the data. The input of this model were the raw 3D accelerometer signals. The Euclidean norms of the 3-axis accelerometer signals were calculated as mentioned in the model above. A moving window STD was used to remove windows with a low STD (i.e., low activity). We used moving windows with a length of 3 s and set the STD threshold to 0.1. The output of this algorithm was a vector of the same size as the accelerometry data in which each sample belonged to a specific state. Each state represented an AR model of the power spectral density (PSD). Then, we computed the total PSD of the gait-related frequencies for each state and selected a threshold that maximized the distinction between gait and non-gait states.

#### 2.3.4. Deep Convolutional Neural Network (DCNN)

Another approach we used for gait detection was a DCNN, a type of artificial neural network, commonly applied in the computer vision and image processing fields. Semantic segmentation is a DCNN task used for detecting different clusters in an image, which assigns a separate label for each pixel according to its cluster. The tri-axial accelerometer data were a matrix with a size 3 × times of the samples; therefore, it can be represented as a two-dimensional image. Then, we used a semantic segmentation model to cluster the data into gait and non-gait segments. Zou et al. [31] used this method for gait detection on data that were recorded by smartphones equipped with an IMU system (Figure 2). They treat the tri-axial accelerometer and gyroscope data as a two-dimensional picture and used the DCNN model to apply semantic segmentation. Their model was inspired by the U-Net semantic segmentation algorithm [51]. In this study, we implemented their algorithm with some modifications. We used only the accelerometer data (without gyro), such that the input of the model was the number of mini-batches × 1 × 3 × 512, where each batch included 64 windows consisting of 512 samples for each of the 3 axes of the accelerometer. Further, we used the batch normalization method [52] for stabilizing the learning process and dropout regularization for preventing overfitting [53].

#### 2.3.5. Long Short-Term Memory (LSTM)

The last model that we used for gait detection was based on LSTM. LSTM cells (Figure 3) are variants of recurrent neural networks (RNNs), which are used for problems of sequential data and can handle long-term dependencies using gate functions [54]. Hence, the final algorithm chosen for the gait detection task was an LSTM network. Each LSTM cell consisted of an input gate, a forget gate, and an output gate, which controlled the important information on the current state to be passed forward and which information was redundant. In bidirectional LSTM architecture, the network is trained simultaneously in both time directions.

The improvement of the bidirectional LSTM is that the current output is not only related to previous information in the time series but also to subsequent information in the time series. It was also shown by Wu, Zhang, and Zong (2016) [56] that adding skip connections to the cell outputs with a gated identity function could improve network performance. Yu Zhao et al. [30] proposed a deep network architecture using residual bidirectional long short-term memory (LSTM) cells for HAR. The dataset that was used in their work was a public domain UCI dataset that consisted of 30 young healthy participant readings of smartphone-embedded accelerometers and a gyroscope placed on the waist of the participants. As in the DCNN model, we applied the Deep-Res-Bidir-LSTM architecture to our datasets with several modifications. The input features were only the three axes of the accelerometer data, and the signals were split into sliding windows at a fixed width of 6 s with an overlap of 50%. The dimensions of the classification task were reduced to a binary classification task (i.e., gait/non-gait). 

### 2.4. Algorithm Evaluations and Statistics

Gait detection algorithms aim to perform a binary classification at any point in time to determine whether a given window is gait or non-gait activity. The performance of these algorithms was evaluated using the following statistics.

True positive (TP) was the number of samples that included gaits that were correctly predicted as gaits. True negative (TN) was the number of samples that did not include gaits that were correctly predicted as non-gaits. False-positive (FP) was the number of samples that did not include gaits that were wrongly predicted as gaits. False-negative (FN) was the number of samples that included gaits that were wrongly predicted as non-gaits. From these statistics, we extracted measures to assess the model’s performance:

Accuracy (Acc) is the percent of correct predictions from the total number of samples:(2)$$\frac{TP+TN}{TP+FN+TN+FP}\times 100$$

Specificity (Sp) represents the ability of the algorithm to correctly predict samples as non-gait among all the non-gait samples:(3)$$\frac{TN}{TN+FP}\times 100$$

Sensitivity (Se) represents the ability of the algorithm to detect the gait samples:(4)$$\frac{TP}{TP+FN}\times 100$$

Precision (Pr) is the percent of correct predicted gait samples from the total number of samples predicted as gait (including non-gait samples that were predicted as gaits (i.e., FP)):(5)$$\frac{TP}{TP+FP}\times 100$$

In addition, we used the receiver operating characteristics (ROC) curve and the precision–recall curve (PR) to visualize the models’ performance. The ROC curve shows the relationship between the true positive rate (i.e., sensitivity) and the false positive rate (i.e., specificity). The PR curve shows the relationship between the precision and the recall (i.e., sensitivity) of the model. In an imbalanced dataset, such as daily living data, the ROC is overly optimistic; thus, the PR curve can be used as an alternative metric to evaluate a model’s performance [57]. We used the area under the curve (AUC) statistic to evaluate the performance in these curves. 

### 2.5. Gait Quantity Measures

To examine the ability of the gait detection models to evaluate gait quantity measures, we computed the correlation between the amount of daily walking according to the lower-back gold standard and according to the wrist-based algorithm. We selected a classification threshold of 0.9 for classifying gait. Only days with at least 8 h of IMU recording were used for each participant. For each day, we calculated the number of minutes spent walking obtained via the lower-back reference value and the wrist-based models. Then, we computed the Pearson correlation between the two measures.

### 2.6. Gait Quality Measures

Four previously described measures of gait quality were extracted from the best wrist-based model and compared to those obtained via the lower-back sensor to begin to evaluate the ability of the wrist-worn measures to characterize the daily living gait of the study participants. The frequency measures included the dominant frequency (Hz) and amplitude of the dominant harmony of the PSD in the locomotor band (0.5–3 Hz) of the acceleration signal. The amplitude is related to the dominance (or strength) of the frequency in the signal [58]. The root mean square (RMS, $m/s^{2}$) quantifies the magnitude of the 3D acceleration signal as a measure of gait intensity [59]. Stride regularity is a measure of the consistency of the walking pattern (higher values reflect greater stride-to-stride consistency, and lower values reflect greater stride-to-stride variability) [60]. 

To increase the certainty that we were extracting gait quality measures from gait bouts and not from false-positive segments, we set a high classification threshold of 0.9. As a result, only samples with a predicted probability greater than 0.9 were classified as gaits, increasing the precision of the model.

## 3. Results

### 3.1. Subject Characteristics

Table 1 summarizes the characteristics of the daily living study participants. Both groups (PD and OAs) had relatively high timed up and go (TUG) scores and gait speeds that are typical of OAs. The Hoehn and Yahr scores indicated that the people with PD generally had moderate disease severity. 

### 3.2. Comparison of the Gait Detection Algorithms’ Performances

Table 2 summarizes the performance of the new anomaly detection method compared to four previously published methods. We assessed the models’ performance on the test set. For the OAs and PD, we averaged the performance metrics of the different folds. 

Figure 4 reports the ROC and PR curves of all four gait detection models. The curves were calculated from the test set predictions of the five-fold cross-validation. Each curve shows the performance of the model in the three study cohorts: PD (blue), OAs (orange), and HYAs (green). The feature-based model’s output was a binary decision (i.e., gait/non-gait) and not a continuous number that represented probability as in the other models. Therefore, the ROC and PR curves are not presented for this model.

### 3.3. Gait Quantity Measured in Daily Living

Figure 5 shows the correlation between the amount of daily walking according to the lower-back gold standard and the activity according to the best algorithm (i.e., DCNN). It shows the correlations in the study cohorts recorded under daily living conditions, i.e., the PD and OA groups. Correlations obtained using the other gait detection algorithms are presented in Table 3 and Table 4.

### 3.4. Gait Quality Measures in Daily Living

Gait quality measures were extracted from the gait bouts that were classified the DCNN model. Group differences in daily living gait quality are summarized in Table 5, Table 6 and Table 7. All the gait quality measures were significantly higher (i.e., better) in the HYAs dataset than in the PD and OA groups as expected, since the PD and OA groups had mobility deficits (recall their TUG times).

In addition, we examined the correlation between the daily quality measures obtained by the gold standard and according to the DCNN model. Table 8 and Table 9 report these correlations.

## 4. Discussion

In this work, we evaluated algorithms for automatic gait detection from wrist-worn sensor accelerometer data collected from OAs with and without PD in real-world, daily living settings. Data from 12 OAs and 18 PD who wore wrist sensors for up to 10 days were used to train and evaluate an unsupervised anomaly detection algorithm and four previously published gait detection methods. All algorithms were first applied to a dataset of HYAs that was collected in a controlled setting to first evaluate and compare the models in a less challenging cohort and conditions. We then applied these five gait detection algorithms to detect gait during daily living in OAs and people with PD.

As presented in Table 2, all models evaluated obtained high performance when applied to the HYA dataset. This indicates that, on some level, they are suitable for the gait detection task. These results also provide a baseline for the performance of the other models, which enabled a standardized comparison between the models. The TUG times of the OA group and the PD group were higher than 13.5 s (recall Table 1). This value is often used as a cut-off point to identify patients with an abnormal gait and an increased risk of falls [44]. This implies that the OAs and people with PD both had mobility deficits. Thus, it was not surprising that the gait detection results obtained for these two groups were not as good as those obtained for the HYAs. This highlights the challenges of gait detection in populations with impaired gait and movement disorders as well as the challenges that are encountered in detecting gait in real-world, daily living settings using a wrist sensor. 

The comparisons between the present results to the previously published findings of the evaluated algorithms were limited and not necessarily direct since some of the performance metrics were lacking in those studies, especially those metrics that were more informative regarding imbalanced data (i.e., precision and recall). For example, Raykov et al. reported only the balanced accuracy (i.e., average of sensitivity and specificity) for gait detection from the wrist sensor, which was higher than the balanced accuracy obtained with our PD dataset. Specifically, we obtained 72.6% in our PD dataset, and they reported 83% with their PD dataset [28]. Their dataset included sensor recordings during uninterrupted and unscripted daily life activities in the participants’ homes and for approximately at least one hour. In our dataset, participants wore the wrist-worn sensor for up to 10 days including during various activities (e.g., driving, sleeping, and cycling) that likely complicated and challenged the gait detection algorithms. In the DCNN paper, only accuracy was reported, and it was 90.22%, which is lower than the accuracy obtained in our experiments (i.e., a 91.7–96.2% accuracy for the different cohorts) [31]. For the LSTM model, the recall and precision reported for walking windows detection were 95.77% and 93.33%, respectively, in healthy adults who wore the sensors on their waist [30]. Several factors can explain the difference in the lower results obtained with our dataset. First, the signal from the wrist sensor is much noisier than the signal from the waist due to the complexities of hand movement. Second, the sensor in their experiment consisted of both accelerometers and gyroscopes which provided more information about the movement and allowed better detection of the activities. Another explanation may be that the classification task was not binary, and the model learned better to distinguish walking from other classes. 

The DCNN approach obtained the highest results among the five methods that we evaluated. This might be anticipated since this is a supervised method in which the labels are provided during the training phase. The relatively low precision obtained among all the models reveals the problem of imbalanced data that are embedded in general gait detection algorithms, and this is even more prominent among OAs and people with PD. Further research that addresses the imbalanced data challenge may help to improve the performance of the algorithms in these populations. 

The anomaly detection-based algorithm achieved high accuracy (99.4%) and relatively high precision (79.1%) in the HYA dataset. This suggests that the anomaly detection approach is feasible for the gait detection task and has the potential to provide a flexible and robust method that does not require labels. When applied to the other datasets, the anomaly detection method also presented comparable results in terms of accuracy, specificity, and sensitivity to the other methods evaluated in the work; however, the precision was lower. The complexity of both the PD and OA gait data (e.g., due to the unrhythmic gait, reduced arm swing) and the non-gait data (i.e., due to the combination of the relatively high duration of stationary activities together with more active ones) challenges the task of density estimation on which this approach is based. More techniques to improve the ability to learn the complicated density, such as modification to the flow architecture to achieve a better semantic structure of the target data, are needed to further explore the possibility of improving the anomaly detection approach [61].

In conclusion, the results of the present study suggest that detecting gait from a wrist-worn accelerometer can be achieved, albeit with some limitations. The imbalance embedded in a daily living dataset was expressed in our results by a significant trade-off between precision and recall. Hence, for identifying all walking bouts during daily living, additional work is needed. Still, for some purposes, the current approaches may already be adequate. For example, if the goal is to evaluate gait quality, it might be sufficient to use a model with high precision (i.e., relatively few false positives) to correctly identify some of the gait segments, even though not all the segments will be identified (i.e., low recall) (recall Table 5, Table 6 and Table 7). In addition, the relatively high correlation between the predicted and gold-standard daily living walking amounts (Figure 5) suggests that we can use this algorithm to estimate gait quantity measures such as daily walking time. We can utilize the ability to extract daily living gait measures from the wrist sensor for more accurate clinical assessment and as a predictor for health outcomes, such as falls [13,14,15,16,17,18].

Nonetheless, it is important to note that, as mentioned in Section 2.2, we considered only walking bouts with a duration of at least 6 s as “gait”, since shorter duration bouts are difficult to reliably detect, in general, but especially using a wrist-worn device, and since most gait quality measures cannot be applied to very short walking bouts [48]. Moreover, several papers that aimed to detect gait from a wrist-worn device also used this approach [27]. However, many shorter walking bouts might be missed in this approach. Future studies should address this limitation by improving the resolution of the detector. Future work should also include more participants to increase the sample size and allow for a more informed comparison of model performance. Different cohorts should also be included to increase model generalizability and to examine the clinical and health value of these daily living gait measures. In addition, it would be interesting to compare the results based on the machine learning methods described here to other approaches that have been used to extract certain aspects of gait from daily living accelerometer data [26,33]. The relatively high compliance that can be obtained with wrist-worn sensors [62] and the present results set the stage for future studies.

## Figures and Tables

**Figure 1 sensors-22-07094-f001:**
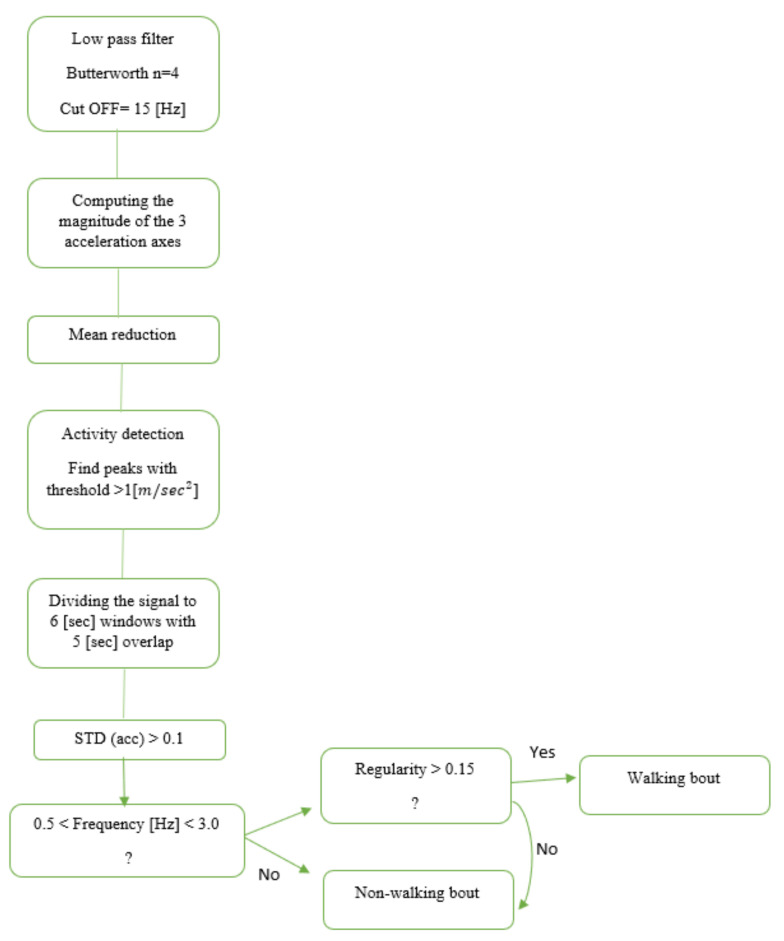
Flowchart based on Keren’s gait detection algorithm [24]. The raw data were divided into six-second windows. The Euclidean norm of the 3-axis accelerometer signal was calculated to minimize any effects of the sensor’s placement and angle or orientation dependency. An activity threshold was set to 1 ($m/s^{2}$) to remove areas in the recording with low activity. Then, each window was classified as a walking bout based on the time and frequency domain features.

**Figure 2 sensors-22-07094-f002:**
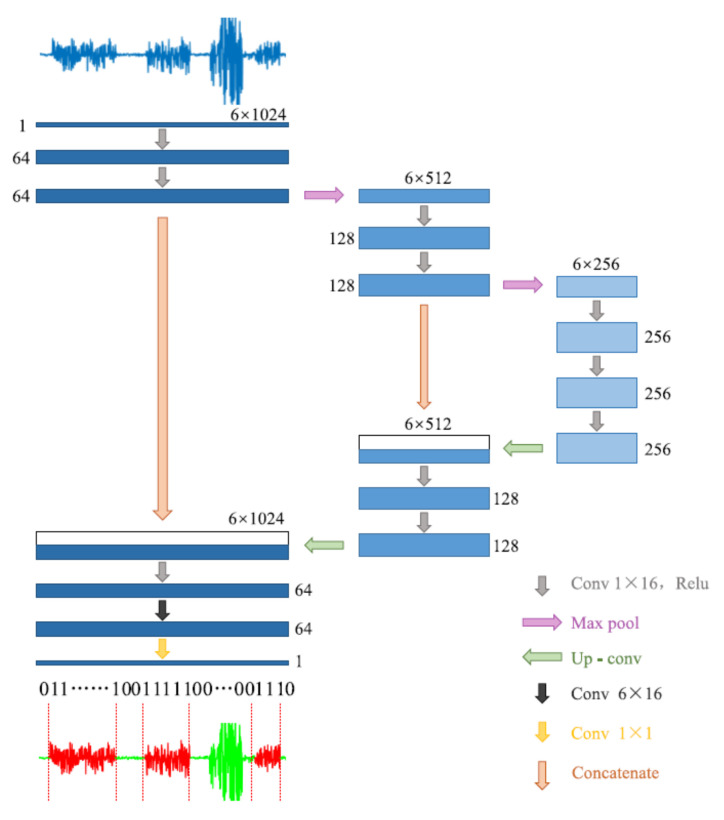
The architecture of the DCNN model. Adapted from Zou et al. (2020) [31].

**Figure 3 sensors-22-07094-f003:**
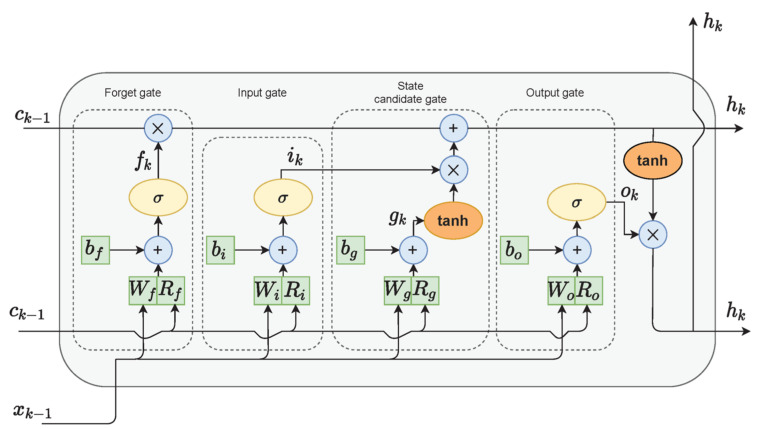
LSTM cell diagram. Adapted from Zarzycky and Ławryńczuk (2021) [55].

**Figure 4 sensors-22-07094-f004:**
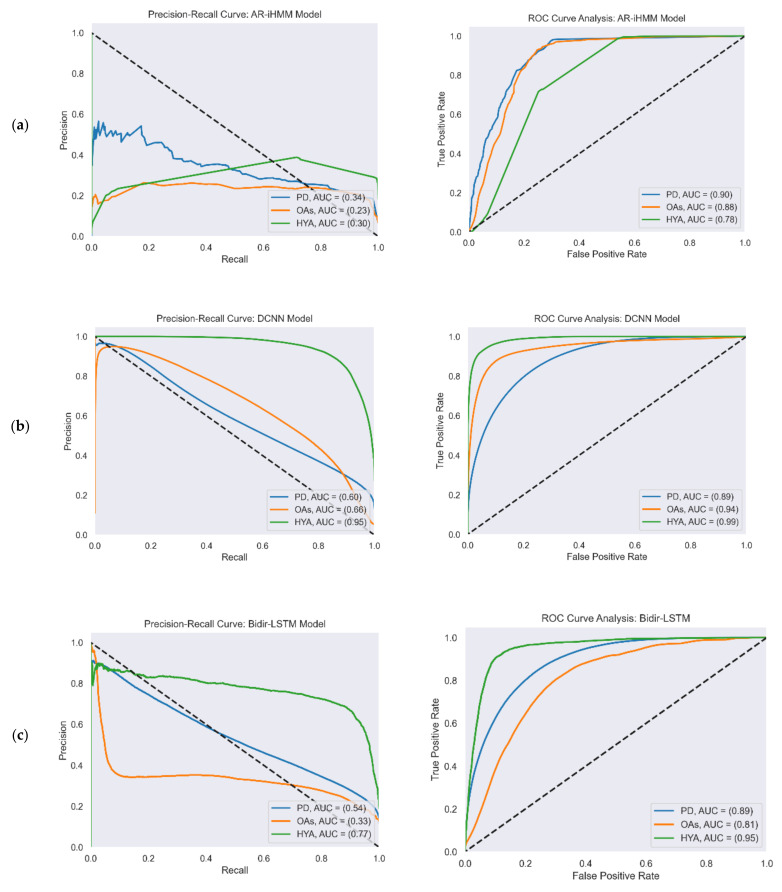
PR (left) and ROC curves (right) illustrate the performance of the four gait detection models with the three study cohorts: PD, OAs, and HYAs: (**a**) PR curve (left) and ROC curve (right) of the AR-iHMM model; (**b**) PR curve (left) and ROC curve (right) of the DCNN model; (**c**) PR curve (left) and ROC curve (right) of the Bidir-LSTM model; (**d**) PR curve (left) and ROC curve (right) of the anomaly detection model.

**Figure 5 sensors-22-07094-f005:**
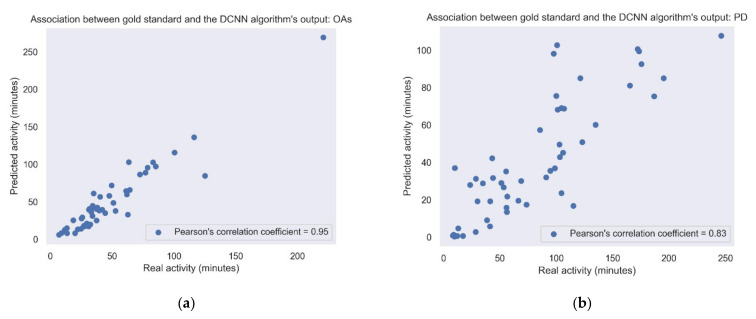
Association between the gold standard and the DCNN model output. Each point represents the sum of the activity for each day. (**a**) Pearson’s correlation coefficient = 0.83; *p*-value < 0.001. (**b**) OAs: Pearson’s correlation coefficient = 0.95; *p*-value < 0.001. When removing the outlier point in the upper right corner, the Pearson’s correlation coefficient changed to 0.9.

**Table 1 sensors-22-07094-t001:** Characteristics of the older adults analyzed in this study.

	No Parkinson’s Disease (No PD)	Parkinson’s Disease (PD)
**No. of Subjects (N)**	12	18
**Age (years)**	76.1 ± 7.3	72.7 ± 8.4
**Gender (M:F)**	3:9	10:8
**Education (years)**	15.8 ± 3.4	15.9 ± 3.1
**Montreal Cognitive Assessment (0–30)**	24.3 ± 5.3	22.4 ± 5.3
**Hoehn and Yahr Stage (1–4)**	-	2.3 ± 0.8
**Disease Duration in Years**	-	5.56 ± 4.05
**Walking aid (No. of Subjects)**	3 (25%)	4 (22%)
**Timed Up and Go (s)**	14.8 ± 3.5	15.1 ± 8.8
**Dual-Task Timed Up and Go (s)**	17.5 ± 5.1	19.5 ± 10.7
**Gait Speed (cm/s)**	99 ± 20	110 ± 27
**Days Sensor Worn**	6.0 ± 2.4	8.1 ± 2.3
**No. of Gait Bouts per Recording**	864 ±548	1264 ± 841
**Average Length of Gait Bouts (s)**	19.5 ± 4.1	20.4 ± 5.4
**Gait Percent (%) per Recording**	7.2 ± 2.6	6.8 ± 4.7

**Table 2 sensors-22-07094-t002:** **Comparing a new anomaly detection algorithm with prior methods.** The algorithm’s performance in gait detection (values in parentheses indicate the STD between the different validation folds).

	Accuracy (%)	Precision (%)	Sensitivity (%)	Specificity (%)
HYAs				
FB	71.0	36.3	77.8	69.5
AR-iHMM	73.2	40.0	94.6	68.4
DCNN	96.0	90.9	83.8	98.4
Bidir-LSTM	90.9	70.6	84.3	92.3
**New Anomaly Detection**	99.4	79.1	44.6	99.4
OAs				
FB	76.2 (2.4)	15.4 (5.8)	59.8 (6.3)	77.3 (2.5)
AR-iHMM	85.2 (8.1)	27.3 (8.6)	78.7 (17.9)	85 (9.43)
DCNN	96.2 (4.2)	61.4 (3.4)	69.8 (9.8)	97.6 (4.9)
Bidir-LSTM	81.7 (6.6)	38.8 (11.9)	81.7 (5.7)	81.8 (7.4)
**New Anomaly Detection**	76.2 (0.5)	29.9 (0.5)	67.6 (0.1)	78.1 (0.6)
PD				
FB	77.6 (5.7)	15.2 (11.3)	57.4 (15.6)	57.4 (5.5)
AR-iHMM	92.6 (4.7)	38.1 (24.0)	50 (23.3)	95.1 (5.0)
DCNN	91.7 (7.4)	40.9 (6.0)	72.9 (7.4)	92.9 (9.2)
Bidir-LSTM	76.6 (4.6)	38.3 (7.1)	88.6 (10.9)	76.5 (6.1)
**New Anomaly Detection**	83.3 (3.8)	20.9 (2.3)	74.7 (8.2)	83.8 (5.4)

**Table 3 sensors-22-07094-t003:** Correlation between the gold standard and the gait detection models in the OAs cohort.

Model	Pearson’s Correlation Coefficient	*p*-Value
FB	0.80	$<0.001{\;}^{*}$
AR-iHMM	0.78	$<0.001{\;}^{*}$
DCNN	0.95	$<0.001{\;}^{*}$
Bidir-LSTM	0.92	$<0.001{\;}^{*}$
New Anomaly Detection	0.84	$<0.001{\;}^{*}$

* Significant after Bonferroni correction ($p=\frac{\alpha }{components};\;\alpha =0.05)$.

**Table 4 sensors-22-07094-t004:** Correlation between the gold standard and the gait detection models in the PD cohort.

Model	Pearson’s Correlation Coefficient	*p*-Value
FB	0.37	$<0.001{\;}^{*}$
AR-iHMM	0.57	$<0.001^{\;*}$
DCNN	0.85	$<0.001{\;}^{*}$
Bidir-LSTM	0.88	$<0.001^{\;*}$
New Anomaly Detection	0.67	$<0.001{\;}^{*}$

* Significant after Bonferroni correction ($p=\frac{\alpha }{components};\;\alpha =0.05)$.

**Table 5 sensors-22-07094-t005:** Daily living gait quality measures in OAs and healthy adults (HYAs).

Daily Living Gait Quality Measure		OAs	HYAs	*p*-Value
Rhythm	Dominant Frequency (Hz)	0.86 ± 0.2	1.53 ± 0.49	$<0.001{\;}^{*}$
Magnitude	RMS ($m/s^{2}$)	1.36 ± 0.25	3.36 ± 0.73	$<0.001{\;}^{*}$
Regularity/Consistency	Stride regularity (unitless)	0.13 ± 0.02	0.47 ± 0.23	$<0.001{\;}^{*}$
	Amplitude dominant Frequency (unitless)	0.43 ± 0.07	0.73 ± 0.22	$<0.001{\;}^{*}$

* Significant after Bonferroni correction ($p=\frac{\alpha }{components};\;\alpha =0.05)$.

**Table 6 sensors-22-07094-t006:** Daily living gait quality measures in people with PD and healthy adults (HYAs).

Daily Living Gait Quality Measure		PD	HYAs	*p*-Value
Rhythm	Dominant Frequency (Hz)	1.21 ± 0.45	1.53 ± 0.49	$<0.001^{*}$
Magnitude	RMS ($m/s^{2}$)	1.16 ± 0.59	3.36 ± 0.73	$<0.001^{*}$
Regularity/Consistency	Stride Regularity (unitless)	0.14 ± 0.05	0.47 ± 0.23	$<0.001^{*}$
	Amplitude Dominant Frequency (unitless)	0.33 ± 0.15	0.73 ± 0.22	$<0.001^{*}$

* Significant after Bonferroni correction ($p=\frac{\alpha }{components};\;\alpha =0.05)$.

**Table 7 sensors-22-07094-t007:** Daily living gait quality measures in OAs compared to people with PD.

Daily Living Gait Quality Measure		OAs	PD	*p*-Value
Rhythm	Dominant Frequency (Hz)	0.86 ± 0.2	1.21 ± 0.45	$<0.001^{*}$
Magnitude	RMS ($m/s^{2}$)	1.36 ± 0.25	1.16 ± 0.59	0.0293
Regularity/Consistency	Stride Regularity (unitless)	0.13 ± 0.02	0.14 ± 0.05	0.1094
	Amplitude Dominant Frequency (unitless)	0.43 ± 0.07	0.33 ± 0.15	$<0.001^{*}$

* Significant after Bonferroni correction ($p=\frac{\alpha }{components};\;\alpha =0.05)$. Entries are reported as the mean ± SD determined for each participant across all walking bouts larger than 10 s. In the HYA dataset, the values used for comparison were the measures from all the walking bouts. In the OA and the PD datasets, we calculated the gait quality measures for each day’s walking bouts. Then, we took the median of these measures for comparison with the HYA dataset.

**Table 8 sensors-22-07094-t008:** Correlations between the gold standard and the DCNN model in the OA cohort.

Daily Living Gait Quality Measure	Pearson’s Correlation Coefficient	*p*-Value
Dominant Frequency (Hz)	0.48	$<0.001^{\;*}$
RMS ($m/s^{2}$)	0.70	$<0.001{\;}^{*}$
Stride Regularity (unitless)	0.48	$<0.001{\;}^{*}$
Amplitude Dominant Frequency (unitless)	0.49	$<0.001{\;}^{*}$

* Significant after Bonferroni correction ($p=\frac{\alpha }{components};\;\alpha =0.05)$.

**Table 9 sensors-22-07094-t009:** Correlations between the gold standard and the DCNN model in the PD cohort.

Daily Living Gait Quality Measure	Pearson’s Correlation Coefficient	*p*-Value
Dominant Frequency (Hz)	0.46	$<0.001{\;}^{*}$
RMS ($m/s^{2}$)	0.48	$<0.001{\;}^{*}$
Stride Regularity (unitless)	0.34	0.016
Amplitude Dominant Frequency (unitless)	0.44	$0.011{\;}^{*}$

* Significant after Bonferroni correction ($p=\frac{\alpha }{components};\;\alpha =0.05)$.

## Data Availability

The data presented in this study are available upon request from the corresponding author and upon consideration of human studies and Helsinki approvals.

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
