# Peer review of "Gait Detection from a Wrist-Worn Sensor Using Machine Learning Methods: A Daily Living Study in Older Adults and People with Parkinson’s Disease"

_sensors, 2022, doi:10.3390/s22187094_

Round 1

Reviewer 1 Report

A new algorithm for measuring gait using a wrist sensor is presented by the authors. In addition to being highly educational, the article also presents interesting findings that may contribute to the continued development of a technique that will make it possible to assess the various gait components in the future using only a sensor on the wrist. I think this study is pertinent and innovative given the need to develop accurate assessment tools with minimal patient collaboration.

Author Response

Thank you for the positive feedback.

Reviewer 2 Report

I believe the wrist-wearing sensor proposed in this study will be an excellent help for remote monitoring and clinical treatment of older adults and patients with various movement disorders. Overall, it was well structured and described according to the style of this journal; however, several modifications are required to improve the quality of this paper.

  1. I recommend improving resolution based on the journal form in Figures 1,2,3,4,5,6; I also recommend improving the form of the tables.

Author Response

Thank you for your helpful suggestions.  We have increased the resolution of the figures to 300 dpi (following the journal’s instructions) and reformatted some of the tables to improve clarity.

Reviewer 3 Report

Thank you for the opportunity to review this interesting manuscript. I very much like the idea of being bold and developing new algorithms instead of just relying with existing deep learning or other solutions. I also like very much, that this was rigorously compared to existing solutions. This here seems to be one of the rare cases where initial ‘negative ‘  results are presented, as the new algorithm is not outperforming the existing. This is really great, as it helps to set a baseline for either further development or stopping there and going to a new methodology.

Please see here my comments and suggestions:

Abstract: written in a misleading way. That needs to be corrected. Yes, there is a new algorithm developed, but this algorithm is not the top performer. The way it is written in the abstract, this is not becoming clear and hence implies that the reader should read the paper because a new algorithm is presented that shows best performance compared with various methodologies.

Section 2.3

Gait detection algorithm and anomaly detection cross-validation methodology:  unclear if  leave-bouts-out or leave-subjects-out or other methods where used --- if subject data is present in both  train and validation data set, that could mean that the models are learning participant characteristics (i.e. overfitting) and be less generalizable. That would be a major shortcoming of the analysis presented here and should be either addressed by correction of methodology or very clearly described in the discussion. 

If leave-subjects-out is used – how balanced are the different ‘folds’ used, as I assume different subjects have different gait behavior and hence some folds might have more gait than others.  

Gait/non-gait – if only >6s bouts are counting as gait, all 1-6s walking segments would be non-gait, which is not true and could have impact on the accuracy. As a huge part of walking bouts are actually happening in less than 6s this leaves a lot of potential walking data annotated as ‘not walking’.  This should at least be mentioned in the discussion.

Results:

A clearer description of the data sets would be helpful – i.e. average number and length of gait bouts per participant,  average number and length of non-gait segments. Otherwise, it is unclear how balanced/unbalanced the data is. Especially as this is later picked up in the discussion and mentioned as a challenge.

Figure 4 – the results from the new algorithm are not shown. Given that a major part of this work is the development and testing of this new algorithm, that is strange. Also in the methods section and the abstract it is mentioned that AUC and AUCPR are calculated for all algorithms. If that is the case, it needs to be part of the main manuscript. Otherwise this needs to be corrected in the methods and abstract.

Section 3.3 would be nice to at least also have the rest of the classifiers as supplement results, not only focus on the DCNN.

How was the amount of daily walking averaged (mean or median)? This could be important as especially PD participants could have  ‘’ outlier bad days’”   or “outlier good days” .

Section 3.4 this section is not adding anything to the topic of the manuscript.  Instead, correlation of the quality measures between wrist and gold standard would have been more interesting similar to section 3.3. Alternatively correlations between these measures and the clinical scores (TUG etc. ) would add interesting information.  Another interesting information would be to see how these statistics differ between the different gait detection methodologies, to understand how much impact this actually has, or if potential sources of error are averaged away by the vast amount of data collected during daily life.  Another alternative would be to compare the OA and the PD, as that is a much more important question compared to a comparison to HYA as done in this section.

Analysis methodologies for sections 3.3 and 3.4 are not described in the methods section

Discussion:  participant numbers are very low. It would be good to also add that to the discussion as a shortcoming. This is especially important as here a new methodology was tested and given the small sample size, it is difficult to really assess how much similar/worse/better it is compared to DCNN. Would be great to compare the same of a 100 participants or more.

Round 2

Reviewer 3 Report

Thank you for including my suggestion.